# Growth Ring Measurements of *Shorea robusta* Reveal Responses to Climatic Variation

**Sony Baral** [1,*]**, Narayan Prasad Gaire** [2]**, Sugam Aryal** [3]**, Mohan Pandey** [4]**, Santosh Rayamajhi** [5] **and Harald Vacik** [1] 

[1] Institute of Silviculture, University of Natural Resources and Life Sciences, 1190 Vienna, Austria; harald.vacik@boku.ac.at

[2] CAS Key Laboratory of Tropical Forest Ecology, Xishuangbanna Tropical Botanical Garden, Chinese Academy of Sciences, Menglun, Mengla 666303, Yunnan, China; narayan@xtbg.ac.cn

[3] Institute of Geography, University of Erlangen-Nürnberg, 91058 Erlangen, Germany; sugam.aryal@fau.de

[4] Society of Natural Resources Conservation and Development, Kathmandu 44600, Nepal; mohanpandey1111@gmail.com

[5] Institute of Forestry, Tribhuvan University, Kathmandu 44613, Nepal; srayamajhi2002@yahoo.com

\* Correspondence: sony.baral@boku.ac.at

**Abstract:** Many tropical species are not yet explored by dendrochronologists. Sal (*Shorea robusta* Gaertn.) is an ecologically important and economically valuable tree species which grows in the southern plains and mid-hills of Nepalese Central Himalayas. Detailed knowledge of growth response of this species provides key information for the forest management. This paper aims to assess the dendroclimatic potential of *Shorea robusta* and to understand climatic effects on its growth. A growth analysis was done by taking 60 stem disc samples that were cut 0.3 m above ground and represented different diameter classes (>10 cm to 50 cm). Samples were collected and analysed following standard dendrochronological procedures. The detailed wood anatomical analysis showed that the wood was diffuse-porous, with the distribution of vessels in the entire ring and growth rings mostly marked with gradual structural changes. The basal area increment (BAI) chronology suggested that the species shows a long-term positive growth trend, possibly favoured by the increasing temperature in the region. The growth-climate relationship indicated that a moist year, with high precipitation in spring (March–May, MAM) and summer (June–September, JJAS), as well as high temperature during winter (November–February) was beneficial for the growth of the species, especially in a young stand. A significant positive relationship was observed between the radial trees increment and the total rainfall in April and the average total rainfall from March to September. Similarly, a significant positive relationship between radial growth and an average temperature in winter (November–January) was noted.

**Keywords:** *Shorea robusta*; community forest; disc; tree-ring; tropical region; Nepal

## 1. Introduction

Dendrochronology is an interdisciplinary tool with wide applications in various sectors. Tree rings could be used to study the factors that affect the earth's ecosystems, forest patch dynamics, forest yield, basal area increment (BAI), annual growth, regeneration, population demography, productivity assessment, or insect outbreaks [1–5]. Dendroecological studies have widely been used in forestry, as they provide answers to several questions related to forest management and resource conservation [2]. However, most of the past tree ring studies have been focused on conifer tree species, and very few studies focus on broadleaved species [6,7]. Tropical species and tropical areas are less explored

globally [6–8]. However, with advancements in scientific equipment and analytical techniques in recent years, tree-ring studies, which were less represented in earlier dendrochronological studies, have expanded into the tropical region and species [9–11]. In South-east Asia, studies of tree rings for the evaluation of growth rates, wood productivity, quality, and rotation cycles have been recorded for a long time [12], but systematic tree-ring research based on the accurate dating of long sequences of growth rings in the Indian region only started at the end of the 1980s [12]. There is very little evidence describing the dendroclimatic potential of tropical species in the South Asian region [9,10,13–15]. However, the most studied tropical tree species from South-east Asia is teak, which can attain hundreds of years in maturity and also produce distinct rings [9,15]. Recent studies using wood-anatomical features to distinguish annual ring boundaries indicate that there are several promising species in the Indian-subcontinent, including Nepal, that can be used to extend the dendrochronological studies of tropical forests [10].

Nepal hosts tropical to alpine climates and vegetation, with ample opportunities for multi-aspect tree-ring studies. In Nepal, the collection of tree cores started at the end of the 1970s, almost at the same time as the initiation of the community forestry programme. However, an institutional study, instituted by the establishment of a lab, started much later [16]. To date, over 80 tree-ring studies covering 22 tree species have been conducted in Nepal; however, most of there were conducted in the temperate and sub-alpine regions [16,17]. Very few tree-ring studies have been carried out on the sub-tropical belt, and none of the studies focused on the tropical region [16–18]. As more than 25 percent of the land area in Nepal is under tropical and sub-tropical belts, it is essential to expand the dendrochronological study in those regions to assess the growth of the trees and also to know their climate sensitivity approaches and responses.

The lowland tropical and subtropical regions of Nepal are almost covered by economically valuable species like *Shorea robusta*, (*S. robusta*) which forms pure as well as mixed forests associated with other species [19,20]. It covers circa 1 million ha, representing about 16% of the total forest area of the country [21,22]. In Nepal, community-based forest management was introduced in late 1970s, which gave priority on the conservation of degraded forests [23]. By doing this, community forests were successful in re-establishing the *S. robusta* tree species over a short period of time [24–26]. However, the management of forests is conservation-oriented and only based on the harvesting of a diameter class greater than 50 cm. The newly emerged *S. robusta*, which dominates forests in Nepal, may need different management practices. In community forests, the steady growth of fuelwood and small-sized poles may be of greater importance for community members than obtaining large-sized timber. However, the maximization of saw-log production has been a traditional interest for state-owned forests. The interest is because of the higher market value of *S. robusta* saw-logs. Recently, the government has prioritized scientific forest management practices, especially in *S. robusta*-dominated forests. In 2014, the Scientific Forest Management Guidelines defined the rotation age of 80 for *S. robusta*; however, there is no scientific evidence on the fixing of rotation age. This is determined by considering that *S. robusta* is a slow growing species. There still exists a knowledge gap with regard to the growth trends in predicting the annual increment of *S. robusta*. The growth and yield models for *S. robusta* forests in the Bhabar, a plain land with sandy and boulder-filled soil adjoining the Churia Hills of Nepal, have been described by [27]. In light of the need for dendrochronological studies, this study (i) investigates the annual radial growth of *S. robusta* from the tropical region of Nepal, (ii) examines potential climatic effects on *S. robusta* growth, and (iii) studies the wood anatomical properties of the species.

## 2. Materials and Methods

### 2.1. Study Species

*Shorea robusta* is a tree species that belongs to the Dipterocarpaceae family and is native to the Indian subcontinent, ranging from Myanmar in the east to Bangladesh, India, and Nepal in the west.

The forest suffered from huge deforestation and degradation in 1970s, so the government of Nepal designated the degraded forest as a community forest for management and subsistence utilization. [28]. It is ecologically and socio-economically very important and a valuable tropical and subtropical tree species in that region. In Nepal, it is found mostly in the Terai region, in the Siwalik Hills (Churia Range), and in mid-hills river valleys. *Shorea robusta* is a moderate-to-slow growing species that can attain a height of 30 to 35 m and a diameter at breast height (DBH) of 2–2.5 m [28]. This species is one of the most preferred and highly valuable tree species for timber production.

## 2.2. Study Area

The study was conducted in the Kankali Community Forest, which is located in Khairani Municipality, Chitwan District (Figure 1), one of the tropical regions of Nepal. The study site was selected based on the existence of multiple forest inventories data [29]. The forest is located at 27.65° N and 84.57° E and covers 749.18 ha. It is divided into five blocks of size ranging from 99.8 ha to 191.44 ha for forest management purposes. The elevation is in the range of 300–900 m. The forests are dominated by *S. robusta,* along with other associated species such as *Semecarpus anacardium*, *Holarrhena pubescens*, *Terminalia alata*, and *Dalbergia sissoo*.

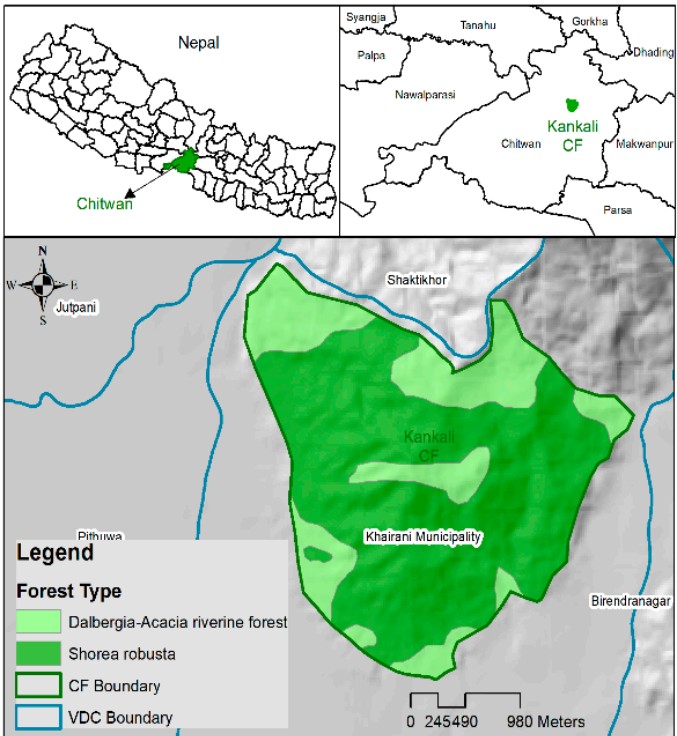

**Figure 1.** Location map of the sampling site in the Chitwan district, Central Nepal.

Climate data analysis from the nearest weather station shows that the area is tropical and monsoon-dominated, with an average annual temperature of 24.4 °C and an annual total precipitation of 2011 mm (Figure 2a). July is the hottest month, and January is the coldest month; the latter comes with frost, which is as an important factor for seedling mortality but also enriches moisture for the trees in the region. About 80% of the annual rainfall occurs during the monsoon season, i.e., June to September (Figure 2a). In the study area, during the past 31 years (1982–2012), there has been an increasing trend in the average annual temperature, while fluctuations in precipitation with decreased rainfall can be observed in recent years (Figure 2b). Monthly trends of temperature and rainfall in Chitwan from 1982 to 2012 C.E are shown in Figure 2; the data show a drier pre-monsoon and autumn climate and an intensively wet monsoon in the study area (Figure 2a,b).

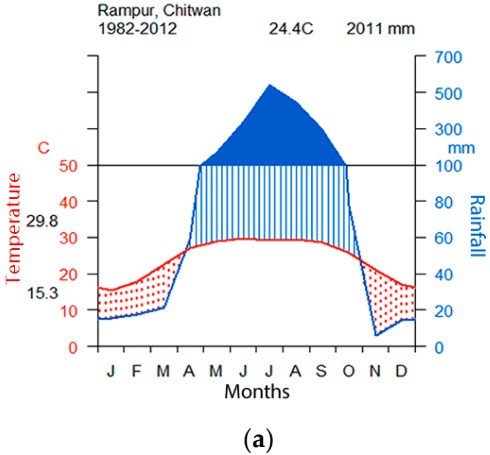

(**a**)

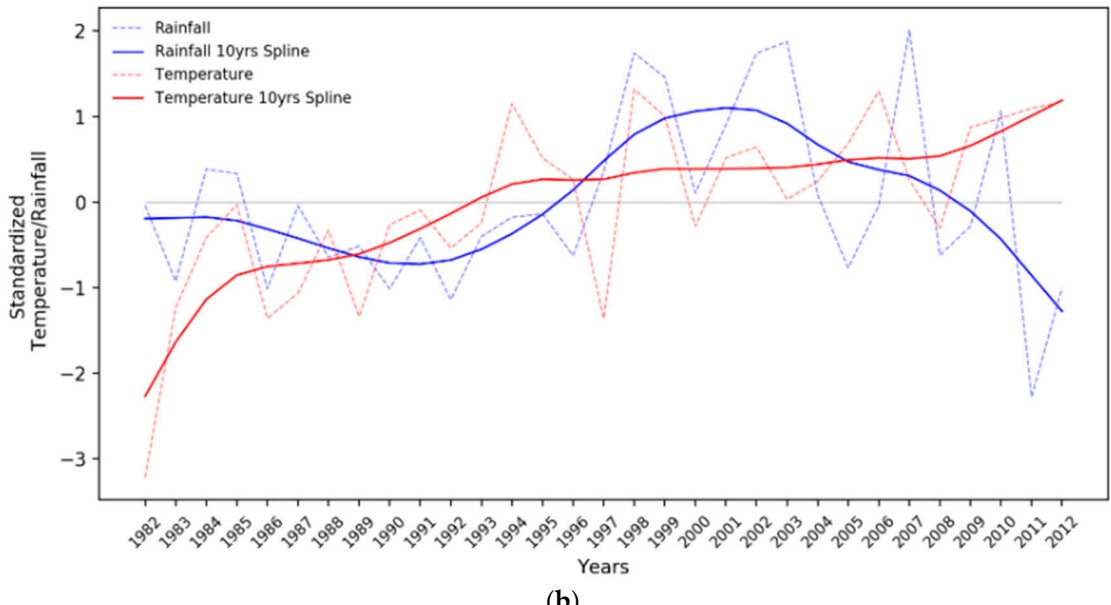

(**b**)

**Figure 2.** (**a**) Monthly data and annual trend; (**b**) trend of annual precipitation and temperature. Data collected from the Rampur meteorological station, Chitwan, Nepal (Source: [30]).

*2.3. Data Collection*

The study site is one of the permanent research and monitoring sites of the ComForM Project of the Institute of Forestry, Tribhuvan University [29], and it is dominated by *S. robusta*. There were no obvious signs of recent disturbances in the study site. For this study, rectangular plots of 20 × 25 m size were laid down where all trees with a DBH of at least 10 cm were measured. Trees with a DBH of 4–9.9 cm were measured within an interior 10 × 15 m plot. Out of the 68 plots laid, a total of 57 plots, which were not disturbed, were selected. Samples for the study were collected in the form of cut-stumps, i.e., a stem disc that was cut at 0.3 m above ground. At least one cut stump per plot was collected from healthy appearing trees. A total of 60 cut-stump samples of *S. robusta* trees were collected, representing all DBH classes from 10 cm to 50 cm in the plot. The samples were brought to the Dendrochronology Laboratory of the Nepal Academy of Science and Technology (NAST) for further analysis in 2017.

*2.4. Laboratory Analysis of Disc Samples*

2.4.1. Preparation of Samples

The samples were processed and analysed following a standard dendrochronological study procedure [1,2]. The samples were left for 15 days for air drying. After air drying, the cross-section surfaces of the cut-stumps were sanded and polished using a belt sanding machine and progressively finer grit sandpaper, such as 120, 220, 320, 400, 600, and 800, until the ring boundaries were visible under the binocular microscope [13]. The fine polishing of the cut-stump surface was done manually using fine grits (1500–2500 grit size) sand papers. Tree-rings were not clearly distinct at the beginning. However, after proper sanding and polishing, they became visible under the stereo zoom microscope.

2.4.2. Counting, Dating, Measurement, and Cross-Dating of Samples

We chose the ring identification criteria or method given for *S. robusta* in a recent study conducted in Bangladesh [10]. Every single ring in each cut stump sample was counted outwards from the pith to the bark using the stereo microscope. Counting was done in four radii in each stump. In most of the cut-stump samples, the pith was not at the centre position. As such, the ring width of tree ring was not symmetrical in all directions, and, in many cases, it was pinched in some locations and missing in those sections. However, with proper observation of the samples under a microscope with necessary rotations in all directions, the counting of each ring was possible. False ring bands also occurred frequently. After counting, the width of each ring was measured at the resolution of 0.01 mm using the LINTAB5 measuring system attached to the PC having TSAP-WIN software [31]. In each cut-stump, the ring-width was measured in the four radii in which counting was done. All 30 bigger sized cut-stump samples which were collected in previous samplings were discarded for further analysis, because most of them were hollow in the center, making it difficult for the accurate dating of the trees. The data of 30 cut stump samples were used for further analysis.

After the measurements was complete, all samples were cross-dated using the alignment plotting technique, looking at the samples, math graph, and cross-dating statistics [31]. The sample showed an irregularity in the ring distribution—basically inclining towards one end—and, as a result, the number of tree rings is not equally distributed in all radii end of the disc. Similarly, the distribution of ring width around the entire disc sample was not symmetrical, i.e., the same rings had some wide sides and other narrow sides. Figure 3 depicts the synchronization patterns of tree growth along four radial directions. The overall growth patterns in each radial direction were found to be synchronized to each other, especially for the years 2003, 2006, 2009, 2013, and 2015. From Figure 3, it is clear that the growth in the perpendicular radial direction tends to synchronize better than the growth in the opposite direction. The alignment plotting technique is a visual method of cross-dating based on the relative scale prepared by matching the width of relatively narrow and wide rings. At first, an average time series of four radii of each cut stump sample was taken, and an average chronology of each tree was prepared. After that, between-tree cross-dating was done using TSAP [31]. The errors in cross-dating were rechecked by using the quality control computer program, COFECHA, and were corrected [32,33].

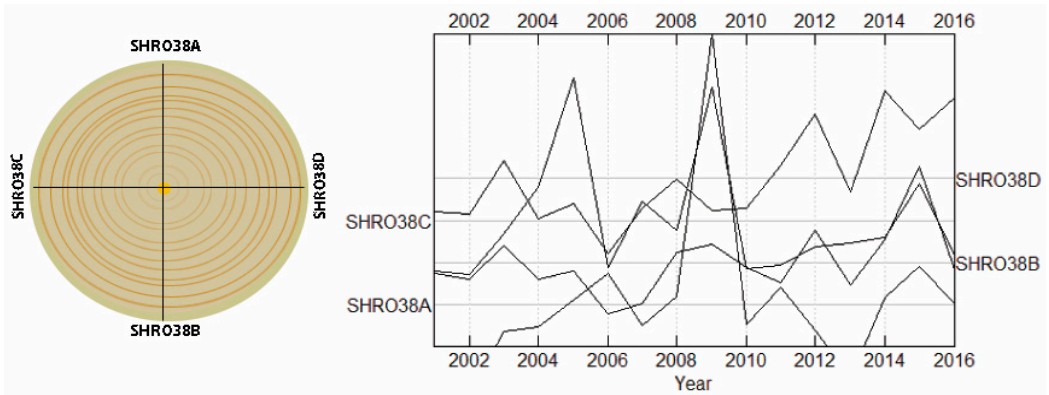

**Figure 3.** Within-tree synchronization of *S. robusta.* The ring-width series annual radial growth (average, maximum, SD) of the sampled trees was calculated based on the cross-dated ring-width data.

### 2.5. Standardization and Chronology Development

A ring-width series incorporates the biological growth trend [1]. Standardization involves removing such age-growth trends or non-climatic effects on growth, which then allows the resultant standardized ring-width of each series to be averaged into a mean value function [34]. The standardized ring-width, also called the ring-width indices (It), is determined by dividing the measured ring widths (Wt) by the estimated widths (Yt), based on a fitted curve.

In this study, standardization was carried out with the computer programs RCSSigFree version 45_v2b [35] and the dplR [36] package based on R software. We used both the modified negative exponential as well as an age-dependent spline curve [37] for standardization and performed further analyses for both chronologies. Finally, the ring-width chronology was developed by averaging the ensemble of detrended tree-ring indices across the series for each year using the arithmetic mean [1]. This produces a mean value function that concentrates on the signal (i.e., climatic information stored by the series) and averages out the noise [34]. Standard residual and signal-free chronology, in which the chronology is prepared by following a signal-free standardization procedure using an age-dependent spline curve in the RCSsigfree program, were thus developed. The major chronological statistics, like mean index, mean sensitivity, standard deviation, autocorrelation, correlation within the tree, correlation between trees and all series, and expresses population signal (EPS), can be computed with the "dplR" package [36] to assess the dendroclimatic potential of the chronologies. Due to short time span of our tree-ring series, we could not calculate Rbar and EPS statistics.

### 2.6. BAI Analysis

The BAI sigmoidal growth model is an appropriate means for detecting changes in tree growth that avoids the detrending and standardizing employed in the calculation of the Ring Width Index (RWI) [38]. First, we averaged four raw ring-width series of each tree to produce an individual tree raw series. Then, we used the individual tree BAI to produce mean an unstandardized BAI series for each year. Ring-width data were converted into tree BAI according to the following standard formula:

$$BAI = \pi \left( R^2_n - R^2_{n-1} \right)$$

where "R" is the radius of the tree, and "n" is the year of tree-ring formation. The BAI series was produced using the "bai.in" function in the "dplR" package [36] in R software [39].

### 2.7. Growth-Climate Relationship

We used tree-ring width chronology to analyse the influence of temperature and precipitation on the radial tree-ring growth of *S. robusta* by using the climatic data from the nearest meteorological station. As the study area is in the tropical region, the response was analysed from January to December of the

current year. Pearson's correlation coefficients were used to quantify relationships between tree-ring chronologies (standard and signal-free) and climate variables. Climatic variables included an average monthly temperature of 12 months, starting from January to December; monthly total precipitation of 12 months, starting from January to December; and seasonal (spring = MAM or March–May; summer = JJAS or June–September; spring–summer = MAMJJAS or March–September; winter = NDJ or November–January) temperature and precipitation of the Rampur station. A correlation analysis was carried out for the time period of 1989–2012 in the cases of temperature and precipitation. For correlation analysis, the statistical package "bootRes" [40], was used based on R software [40].

## 2.8. Microscopic Observations of the Thin-Section of the Disc Sample

In order to assess the wood anatomical properties of *S. robusta*, a thin section was prepared using core microtome (between 15 and 30 μm) by following the procedures described in [41]. Small blocks of wood samples were boiled in water for about five hours to soften the wood. After that, 25 μm thick cross-sections were sectioned using a GLS1 microtome [42] at the Dendrochronology Laboratory. Permanent slides were prepared and photographed using a digital camera under a light microscope. Anatomical terms were used following the International Association of Wood Anatomists (IAWA) guidelines [43].

## 3. Results

### 3.1. Annual Growth and Dendroclimatic Potential of S. robusta

Of the 60 sampled cut stumps from 10 cm to 50 cm diameter-sized trees, the 20–30 cm diameter class was the most frequent class. Our result is based on 30 trees in the 20–30 cm DBH class, because bigger trees (>30 cm DBH—some up to 51 years old) were discarded due to their rotten pith or hollowness in the inner portion of the stump, making counting the true age of those trees difficult. Therefore, in the final data set, there were, altogether, 30 tree-ring width series, with an average of four radii data in each series. The mean age of the sampled trees was 22 years (Median = 23 years, Max = 28), while the mean DBH of trees was 16.5 cm. The average radial growth of *S. robusta* was 3.173 mm per year (Max = 8.88 mm, SD = 0.477).

There was a total of 120 raw tree-ring width series prepared from 30 cut-stump samples, each measured in four radii directions (Figure 3). Then, the four radii of each tree were averaged, and the value was used as one series. Therefore, we have a total of 30 tree-ring width series. The mean raw chronology of 30 tree-ring series is depicted in Figure 4, along with a number of samples representing the chronology for each year. The raw chronology shows that the annual radial growth was slow in the beginning, then it rapidly increased, and then slightly decreased in the years 2009 to 2016. The raw chronology in Figure 4 may have some external and internal influences on growth. Figure 5 presents standard and signal-free chronology. Both types of chronologies show an increase in growth in the earlier period and a slight decrease in the latter period (Figure 5).

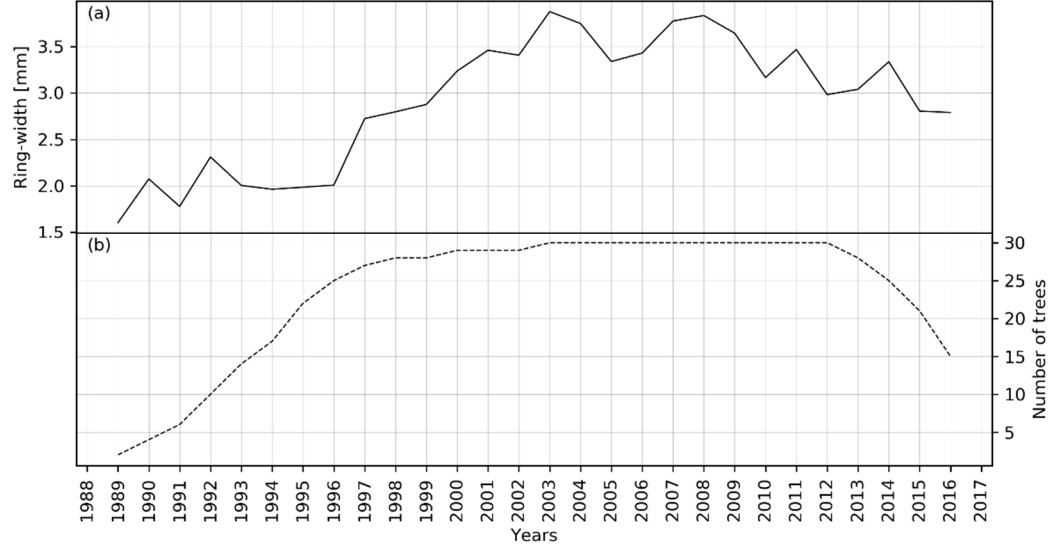

**Figure 4.** Figure showing the raw mean ring-width chronology (**a**) of *S. robusta* from Central Nepal and number of trees used for building this chronology (**b**).

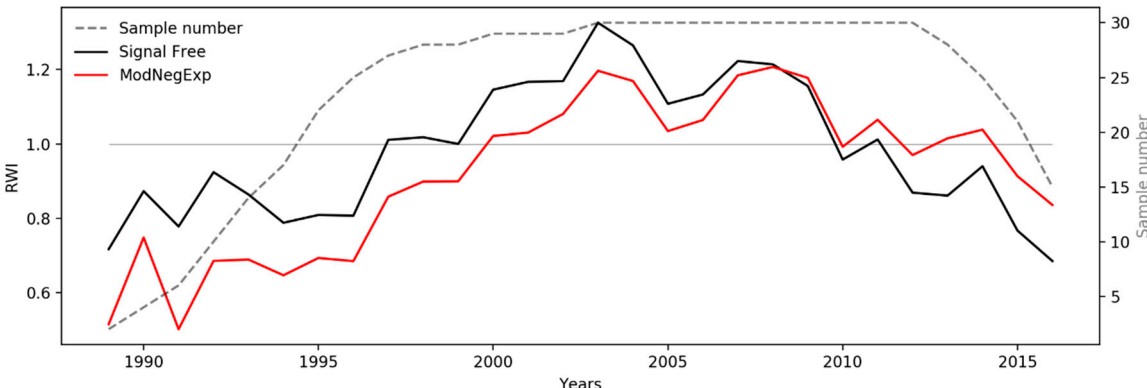

**Figure 5.** Standard ring-width chronology of *S. robusta* from Central Nepal. This indicates the change in growth trend from positive to negative since 2003 CE.

The chronology statistics revealed a dendroclimatic potential of *S. robusta* with moderate mean inter-series correlation (0.409) and mean sensitivity (0.264). Since the chronologies were short and the tree-rings were asymmetric, i.e., the position of pith was not exactly at the center of the disc, this created some problems in synchronization among the series, lowering the inter-series correlation. Figure 6 reveals associations between the different size variables of *S. robusta*. Results indicated that the DBH and age of the sampled trees were weakly correlated, while the DBH and height were strongly correlated to each other.

There is a positive relationship between the DBH and height of trees. Radial growth increases with tree age, but there is not strong relationship between growth and other variables.

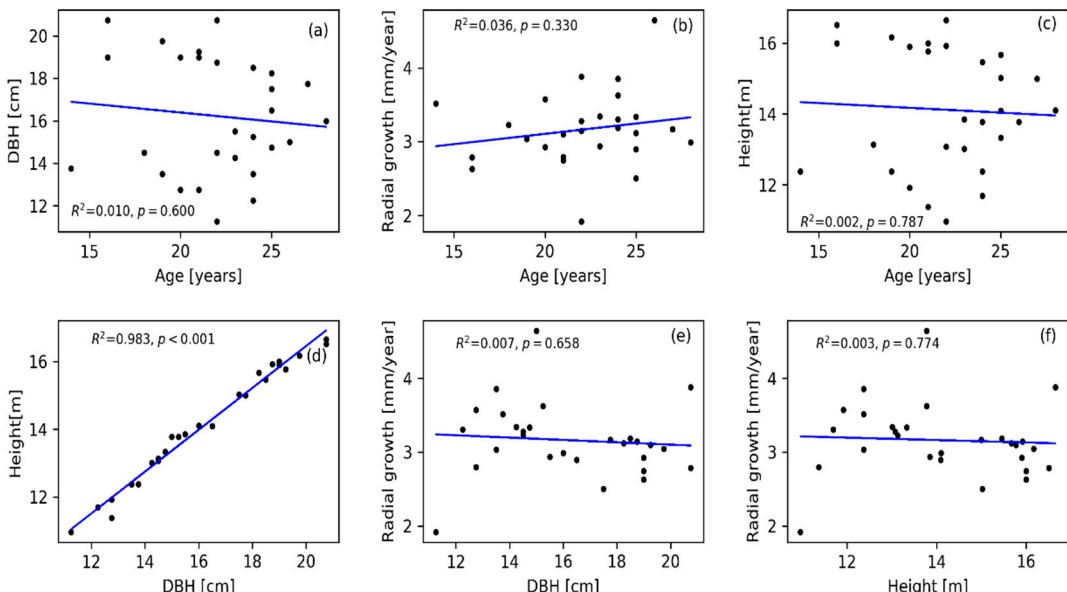

**Figure 6.** Relationship between the growth and different size variables of *S. robusta*; (**a**) DBH-age relationship; (**b**) age-growth relationship; (**c**) age-height relationship; (**d**) DBH-height relationship; (**e**) DBH-growth; and (**f**) height-radial growth relationship. DBH: Diameter at breast height.

### 3.2. Growth-Climate Relationship of S. robusta

The analysis of the climate-growth relationship indicated that growth of *S. robusta* is generally correlated with temperature (Figure 7). There was a significant positive correlation between the growth of *S. robusta* and temperature during the winter months and winter season (December–January–February, DJF) and a significant negative correlation with the temperature in May (Figure 7). Similarly, rainfall also favours growth, with a positive relationship with precipitation in most of the months (Figure 8).

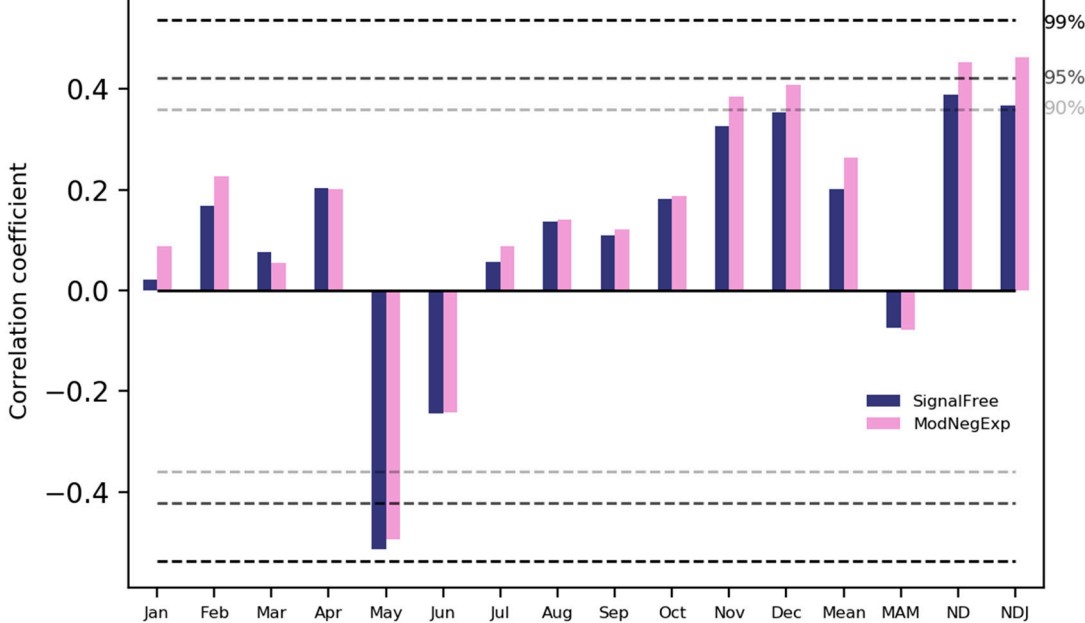

**Figure 7.** Correlation of *S. robusta* ring-width chronology with temperature. Jan, Feb, ..., Dec are months. Mean, MAM, ND, and NDJ indicate the annual average temperature, the average temperature of March to May, the average temperature of November–December, and the average temperature of November–January, respectively.

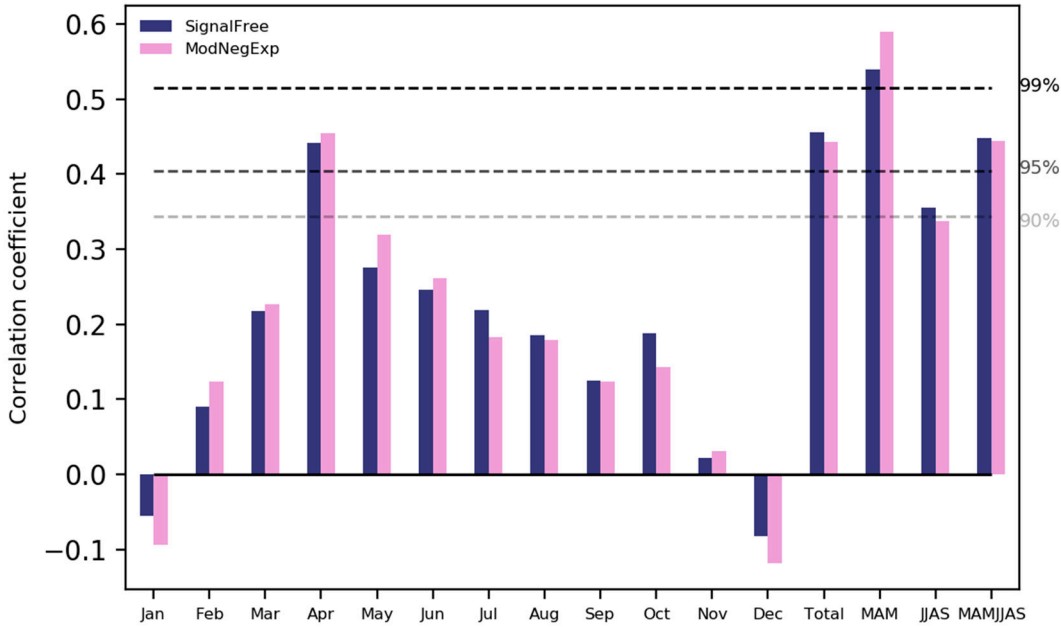

**Figure 8.** Correlation of *S. robusta* chronology with precipitation. Jan, Feb, ..., Dec are months; MAM, JJAS, and MAMJJAS indicate average total precipitation of March to May, June to September, and March to September, respectively.

### 3.3. Basal Area Increment (BAI) Analysis

Figure 9 shows the BAI of the sampled trees with time. The BAI was seen to increase in the early ages of trees until nearly 2009. Afterwards, it was found to decrease slightly.

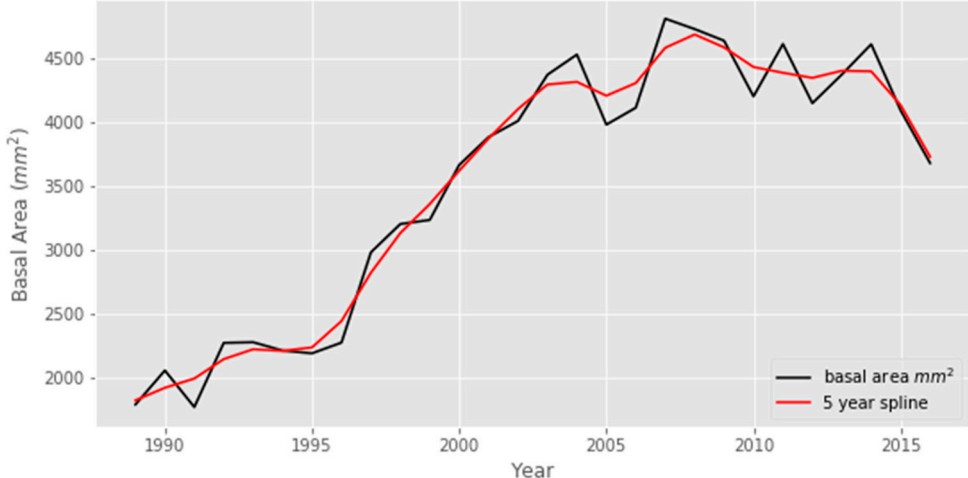

**Figure 9.** Basal area increment (BAI) of *S. robusta*.

The basal area measured by [44] in different time intervals, between 2005 and 2016, shows a positive incremental change throughout the period, with almost double increment in an eleven-year period from 2005 to 2016. Removal is similar in period I and period II; however, it decreases in period III (Table 1).

**Table 1.** The total basal area of *S. robusta* in different years of measurement based on permanent sample plots.

| Year | Basal Area (square meter/ha) | Period | Removal m³/ha/year |
|------|------------------------------|--------|--------------------|
| **2005** | 20.3 | I (2005–2010) | 3.26 |
| **2010** | 28.9 | II (2010–2013) | 3.65 |
| **2013** | 33.2 | III (2013–2016) | 0.13 |
| **2016** | 37.1 | | |

Source: Data from 2005, 2010, and 2013 were obtained from the IoF ComForM project.

### 3.4. Wood Anatomical Analysis

The microscopic features of *S. robusta* wood in the transverse section were analysed (Figure 10). The detailed wood anatomical analysis shows that the wood of *S. robusta* was diffuse-porous with growth rings boundaries that were either vague with marked gradual structural changes or not separable in normal magnification (Figure 11). Vessels were evenly distributed, solitary, or in radial multiples of 2–3 (Figure 11). Solitary pores were round and oval in the cross-section, while the thick walled fibres were mostly polygonal and sometimes round in outline.

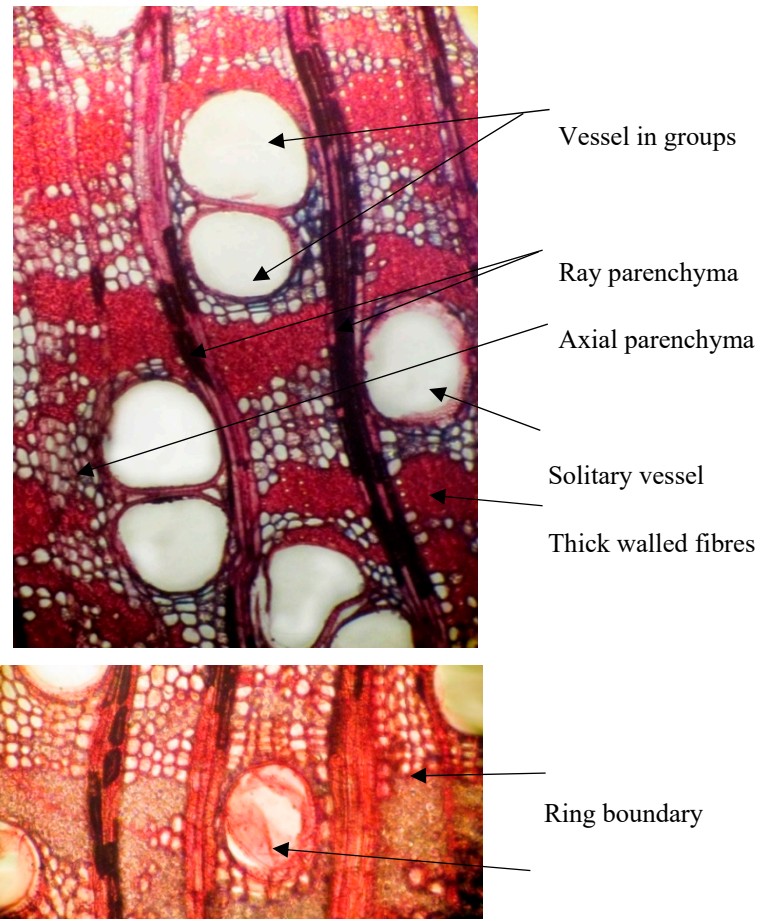

**Figure 10.** Showing a transverse section of *S. robusta* wood.

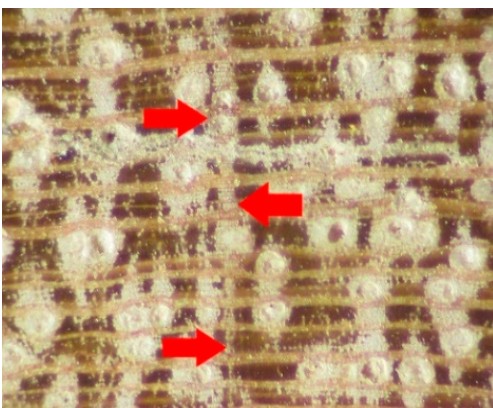

**Figure 11.** Growth ring analysis showing ring boundary in low magnification.

## 4. Discussion

### 4.1. Growth Depends on Various Underlying Factors

Growth depends on several biotic and abiotic factors like size, age, soil condition, nutrient availability, water regime, temperature, light, competition, and fire. [1,14,45]. It also depends on the type of management that considers the time of thinning and harvesting. *S. robusta* is a moderate-to-slow-growing species [28]. The average annual radial growth observed in the present study is similar to that observed for other tropical tree species from the Indian sub-continent [13–15,46] and non-tropical broad-leaved species from adjacent regions [4,43–52]. The growth of *Tectona grandis*, *Cedrela toona*, *Michelia champaca*, and *M. nilgirica* ranged from 1.0 to 4.1 mm per year in different regions in India [13], whereas the average annual radial growth of the tropical teak tree was 2.15 mm and 3.10 mm in Dadeli and Shimoga, respectively, of the Western Ghat region in India [15]. The diameter increments of young individuals of various eight tropical tree species (*Acacia catechu*, *Albizia lebbeck*, *Azadirachta indica*, *Dalbergia sissoo*, *Gmelina arborea*, *Millettia pinnata*, *Tectona grandis*, and *Terminalia bellerica*) in the coal mine spoil site in the tropical region of India were higher than those of the present studied species and ranged from 10.1 to 28.9 mm per year [46]. The average growth rates of several tropical species such as *Shorea wallichiana* in the tropical dry deciduous forest at Mudumalai, Southern India, was 0.1 to 8.39 mm per year [14], and the growth varied with time and species.

The growth pattern is in line with the natural biological growth trend [1,2], as well as the likely influence of the ongoing climate change in the area. Under ideal or normal growth conditions, a tree grows slowly during the juvenile or seedling stage, followed by rapid growth in the young stage, and then reaches its maximum growth rate with a negative exponential continuous decrease following, which is maintained until the end of the tree's life [1,2]. This is because *S. robusta*, in the observed forest stands, is competing with neighbouring trees due to the absence of thinning activities. The low average radial growth and the decreasing growth in recent years is a consequence. Selective thinning practices may lead to the development of the crown, thereby allowing the trees to reach the target diameter much earlier [20,26]. In many studies, the lack of targeted thinning practices has been observed [23].

The chronology statistics (mean sensitivity and inter-series correlation) observed in the present study are within the range of those reported for the different conifer and broadleaved species from Nepal [3,4,51–56]. However, the length of the chronology developed in the present study is very short compared to that of previously prepared chronologies from Nepal [3,4,51–56]. Cook and Kairiukstis [34], studies the relationship between height and DBH with growth found that DBH is a weak indicator of tree age.

### 4.2. Climatic Factors Mainstream Growth

There was a significant positive relationship between the growth and precipitation in the month of April and in the spring (March–April–May, MAM), as well as between the summer period (June

to September) and the total annual precipitation. From the growth-climate response analysis, it is clear that precipitation in the spring season is the main limiting factor for the growth of *S. robusta*. The negative relationship of growth with temperature in May indicates that temperature-induced moisture stress during the late spring season is critical for the growth of the species. During the spring season, the temperature increases rapidly, and when trees suffer from drought stress, growth is limited. The importance of the precipitation observed in the present study is also reported in studies from the middle to high mountains in Nepal [52,57,58] and some species in tropical areas of the Indian sub-continent [15,58,59] and Thailand [60]. Temperature-induced moisture stress with a downward growth trend during the recent few years has been seen in both standard and signal-free chronology. This could be due to the temperature-induced moisture stress or drought stress in the area, as there is a continuous increase in temperature but a decreasing trend in precipitation observed in recent years in the study area (Figure 2). A similar positive relationship between annual rainfall and the growth of tropical teak trees was also found in Western Ghat [15].

### 4.3. Basal Area Increments for Growth Analysis

This can be related to the effects of forest management practices and weather conditions. On the one hand, low-intensity thinning is causing high stem numbers, which can lead to a decrease in BAI with time. On the other hand, the long-term BAI increase could be related to the biological growth of young trees and is also favoured by the temperature increase in the study region during recent decades. As the study areas lie in the humid region of Nepal, with an average annual total rainfall of 2011 mm, an increase in temperature may benefit the growth of *S. robusta*. The recent slow increment or decrease in BAI could be due to moisture stress in recent years. BAI consistently increased for all trees of different age classes in this study. Our BAI chronology showed an increasing trend in the curve, indicating that the forest stand in the study area is still in the growing stage. *S. robusta* can reach a DBH of up to 2.5 m during its maturity, and it can gain a height of up to 35 m [61]. In our study site, the average (maximum) DBH of the analysed sampled trees was 16.5 [23] cm, while the average height was 14 m. Similarly, the oldest recorded individual was 28 years old. The size class (age and DBH) distribution of the studied trees shows the current study forest stand to be still young. The increase in growth in young trees is expected from the sigmoidal model, because BAI should increase as young trees produce an increasingly larger leaf canopy [62]. A constant raw ring-width over time means that the tree is producing an increasingly larger amount of wood each year. This trend is not indicated in the ring width index (RWI) because of standardization or detrending of chronologies [63]. Generally, BAI may continue to increase [64,65] or stabilize [64] in healthy stands, but it only shows a decreasing trend when trees begin to senesce or are subject to significant growth stress [5,64,66,67]. This finding is supported by the results from the analysis of the BAI from tree rings; however, management intervention effects in the same community managed forest was also observed in the *S. robusta* forest [23,44]. The appropriate management with the presence of different age classes is more important for the production of commercial saw-log.

### 4.4. Wood Anatomical Analysis as an Indicator

Tyloses were common, partially or completely blocking the vessel lumen. The wood was composed of axial and ray parenchyma cells. Axial parenchyma was paratracheal and associated with vessels and vascular tracheids; it was sometimes vasicentric, confluent to aliform, marginal or seemingly marginal at the end of the growth rings, and abundant in bands more than three cells wide. Resin canals gum canals were present in tangential lines, giving growth ring-like formations, fairly smaller than vessels, and round-to-oval in shape.

The growth rings of *S. robusta* were indistinct in cross-sections but could be identified after sanding and polishing. They could also be identified from the wood anatomical study. In the current study, it seems that marginal axial parenchyma cells in bands are forming growth rings. However, the continuous formation of growth rings was not observed in the cross-section. Sometimes,

the irregular parenchyma bands formed during xylem increment in *S. robusta* can be mistaken for annual rings [68]. In some cases, the tangential arrangement of traumatic gum canals was causing the formation of ring-like growth structures. Therefore, the resin canals of *S. robusta*, which is hardwood and from lower elevation range, may not be annual rings because tropical trees often produce more than one growth rings per year, and the feature differs from opposite sides of the same tree [68,69]. Previous studies also found similar features in *S. robusta* wood. The darkening of growth rings was also observed in the *S. robusta* wood collected from the Terai and mid-hills of Nepal, but the growth rings were not annual rings [70].

Only limited species form true annual rings in tropics due to a lack of climatic extremes that alter cambial growth. In addition, site-specific and endogenous factors play a vital role in wood development, which makes it difficult to understand and interpret the tree-ring features in tropical trees [71]. Large vessels (about 50 μm in diameter) in *S. robusta* seem to be adapted to enhance water conduction, as large vessels conduct water more efficiently. In general, vessel diameter, rather than vessel number, determines the potential hydraulic conductivity [72]. A thick band of vascular tracheid and fibres observed in *S. robusta* wood are a characteristic feature in a climax species [73] and are directly linked to the mechanical support function.

## 5. Conclusions

In the wood anatomical analysis, the growth rings of *S. robusta* show that the wood is diffuse-porous. *S. robusta* is generally a difficult species to cross-date due to asymmetric ring patterns the prevalence of diffuse, false, and missing rings; however, dating and measurements of annual growth ring of this species are possible and feasible. The raw tree-ring width chronology and BAI showed that the studied *S. robusta* forest is still young and is in the growing stage. Moisture stress during spring and summer seasons or year-round is the main limiting climatic factor for the growth of the species. The results show that diameter growth was influenced considerably by rainfall during the growing season, as well as by high temperature. In addition to the annual ring-width variables, the continuous monitoring of growth was done using bands or a digital-dendrometer. The wood-anatomical features in micro-cores, were found to be useful for the quantitative measurement of growth and growth dynamics at the seasonal, sub-annual, and annual level. This method could also be used to monitor the multi-proxy response of growth of *S. robusta* or other tropical species to climate change and variability. Based on our experience that tree-rings around the trunk in *S. robusta* are not symmetric, collecting disc samples instead of tree cores as much as possible or collecting more than two cores per tree will ensure the accurate dating and measurement of annual radial growth. The successful dating and measurement of tree-rings in one of the important tropical species in the present study showed that there is a huge potential for dendrochronological research in the tropical regions to fulfill the gaps in forestry-related issues, including calculating forest stand age, forest patch dynamics, forest's annual growth, carbon sequestration, the sensitivity of growth to climate change and variability, and the understanding of the long-term climate of the past. The scientific forest management guidelines for the *S. robusta*-dominated forests in Nepal determine the harvesting cycle of the forest. In community forests, the production of fuelwood and small-sized poles may be of greater importance for community members than obtaining large-sized saw-logs. In state-owned forests, saw-log production requires different management practices in order to increase the average radial growth of the target species. Finding the appropriate age is most important factor for determining tree harvesting.

**Author Contributions:** S.B., N.P.G., S.A. and H.V. worked jointly on the study design including framing of the manuscripts. S.B. collected data from the field. N.P.G., S.A. and S.B. carried out the lab analysis of tree rings. M.P. did wood anatomical analysis. S.A. did analysis of lab data. S.B., N.P.G. and S.A. jointly wrote the paper and H.V. and S.R. worked in improving the paper quality.

**Funding:** This research was funded by the APPEAR—Austrian Partnership Programme in Higher Education and Research for Development funded under 'Austrian Agency for International Cooperation in Education and Research (OeAD-GmbH) and the APC was funded by the OA publishing fund at BOKU library services.

**Acknowledgments:** We are thankful for the APPEAR—Austrian Partnership Programme in Higher Education and Research for Development funded under 'Austrian Agency for International Cooperation in Education and Research (OeAD-GmbH) for the financial support.' We are also thankful to the Institute of Forestry, Pokhara, Nepal for providing time series data of 2005, 2010 and 2013. We are also grateful to the forest officials and community forest user groups' member of Kankali community forest for genuinely sharing information without which this research would not be possible. We extend our thanks to enumerators and field research facilitators for their support. We are grateful to two anonymous reviewer and editors for their valuable comments to improve the earlier version of this manuscript.

**Conflicts of Interest:** The authors declare no conflict of interest.

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
