# Peer review of "Growth Ring Measurements of Shorea robusta Reveal Responses to Climatic Variation"

_forests, doi:10.3390/f10060466_

Round 1

Reviewer 1 Report

You have some useful and interesting data on tree growth patterns.  However, the current manuscript has almost no general context and thus is more appropriate for a regional than for an international journal.  I think that putting the study in a more general context is feasible, but this will require a substantial effort.  Also I have a number of other major concerns such as some procedures (e.g., sample selection) need to be more fully described,  and there is some confusion or inconsistency in the presentation of results, which must be addressed.  Below I indicate my main points followed by a list of specific comments.

Main points:

1.  The Introduction needs to be rewritten.  This section starts directly in with the species and the system.  However, for an international journal there needs to be an introduction to the general questions being addressed.  Otherwise the paper would be more appropriate in a regional journal.  The first two paragraphs of the Introduction would be better as part of the study area and species description; they could be incorporated into section 2.1.  The Introduction needs to start with a part on the relevance of the work to an international audience.  What are the issues of general relevance that are being addressed?

2.  In the section on data collection (lines 161-171), it is essential that more information be given on sample selection.  How were trees chosen for sampling?  How many plots were used and how many trees were selected from each plot?  What was the stand structure and history of the forests in the plots?  Was there evidence of recent disturbance?  This kind of information is critical to interpreting patterns of growth of individual trees. 

3.  Given that you are working with a tropical species, it seems likely that rings could be indistinct and not necessarily indicate annual increments.  Thus it is important to say something about the distinctness of rings and why you think they indicate annual increments.  Was there any evidence of false or missing rings, and are these likely to occur in this species.  These characteristics of the rings need to be described early on in the Methods.  They could be described at the beginning of section 2.3.2 where you address counting and measuring the rings. 

4.  The trees sample all appear to have been very young, which is an important consideration for evaluating the results, especially since I assume this species can get very old, at least as indicated by the large size it can attain.  Early in the results there should be a good indication of the age of the trees used.  Sizes should also be clearly indicated as I am confused about the size range because it is indicated that the sizes are 10-50 cm DBH but in Figure 6 the largest tree is only about 24 cm DBH, and most trees are less than 20 cm DBH.  These inconsistencies need to be clarified.  Also the sample sizes need to be clearly indicated for all results.

5.  Throughout the reporting of the results, there needs to be more care in clearly describing the results, especially in relation to the figures and tables.  For example on line 324 it is indicated that DBH and height are strongly correlated, but the panel on figure 6 shows no relationship at all between these variables.  The text of the results needs to be very carefully checked against what is actually presented in figures and table.  Also the figures need to be more clearly presented.  For example the axes labels on Figure 6 are way too small and the letters (a, b, and c) indicated in the legend are not visible on the figure.

6.  There needs to be clarification of the nature of growth rings in this species and whether annual rings can be accurately determined and measured.  There are substantial inconsistencies in section 3 (Results and Discussion) concerning this issue.  For example, the statement on lines 465-468 would indicate much of the parts presented earlier, which are based on precise measurements of annual rings, are questionable.  These inconsistencies need to be fully explained and resolved.  As indicate in my third point above, you need to be very clear in addressing this issue early in the methods.  It is relevant to most of the analyses in the manuscript.

7.   A critical concern is that no general discussion or conclusions are presented that would be of much relevance to an international audience.  Just as with the Introduction, in the last sections of the manuscript the study is placed only in the context of the study system and of the region where the study occurred.  It is essential that the work be placed in the context of concepts related to tree growth patterns and dendrochronology at a more general level.  I think that this is feasible, but has not been done.  Otherwise, the paper should be directed to a regional journal.    

Specific comments and suggestions for wording changes:

Lines 26-29.  This sentence is confusing and needs to be reworded.  

Lines 29-30.  This sentence does not make sense without context provided.  Does this apply to individual trees or stands, and trees of what sizes and growing in what types of conditions?  These points are clearer later, but not here in the abstract.  It might be best to simply remove this sentence.    

Line 35.  “the season” is unclear.  Which season?  Also this sentence seems to be rather redundant with the following sentence and perhaps is not necessary.     

Line 56.  I do not understand the meaning of “from east to west”.    

Lines 62-64.  Does this mean forests were the species is greater than 50% of the trees? 

Lines 69-71.  This part needs clarification.  Did these “naturally emerging” forests not have S. robusta before or do you mean the forests have a different structure because of altered management? 

Lines 80-84.  This part is confusing and seems out of place being attached to the end of a paragraph that mostly covers a different topic. 

Line 110.  What does “appropriately” mean?  Do you mean correctly or accurately?    

Lines 114-117.  This sentence is awkward and needs to be reworded.

Lines 126-128.  This sentence is not necessary and can be deleted.

Lines 153-154.  The last part of this sentence is confusing.  What does “seedling decimation” mean and how does frost enrich moisture?

Line 174.  I suspect that “processed analyzed” should be “processed and analyzed”.

Lines 192-193.  How many samples were discarded?  I assume this reduces the number of samples used from the 60 indicated in section 2.2.  The final number of samples (trees?) used should be indicated here.

Lines 206-207.  This sentence is confusing.  What does “noise” mean here?

Line 241.  You need to clearly indicate what specific variables were used, especially what seasonal variables.  How many variables in total were used?  I suspect this is quite a few variables so was any adjustment made for significance of the correlation in relation to multiple comparisons?

Line 256.  Were there still 60 samples after some were discarded, as indicated on lines 192-193?

Lines 257-260.  This part is confusing.  Are you indicating that there are missing or false rings, and do you mean by “synchronization” the number or the width of rings on different radii?  I assume width, but this needs to be clear here.

Lines 270-271.  Does this mean that only 30 of the 60 trees were used?  Also were there any trees older than 28 years that were not included in the 30 trees?  What was the average age of the trees used?

Line 378.  How are young and mature age classes defined?  Because the oldest trees appear to be about 24 years old (Figure 6) can any of these trees have already reached a mature age class?

Lines 379-380.  You need to be cautious in extrapolating from individual tree to stand-level growth.  These can be very different, depending on stand characteristics such as tree size structure.  This distinction needs to be considered here and throughout the rest of this paragraph.

Line 400-404.  Global change could be important for growth of your trees, but you have no data to confirm this as young trees could be expected to grow rapidly in any case.  Thus caution in interpretation is warranted.

Lines 420-421.  This seems inconsistent with the indications above that growth rings were distinct and could be clearly measured.  This inconsistency is critical and must be explained.

Line 486-488.  This type of comparison of the difficulty of determining growth rings in young versus older trees is not described earlier.  This comparison should not be in the Conclusions section if it is not fully described in the methods and results.

Lines 491-493.  It is unclear what “enhanced” means here as this type of trend would be the normal expectation for young trees. 

Author Response

Manuscript: forests- 489388

Title: Dendroclimatological growth response of Shorea robusta using growth ring measurements

Thank you for providing the valuable suggestions and the opportunity to re-submit our manuscript. We thank both reviewers for their comments and suggestions. We believe that the following statements and the related changes in the manuscript would adequately address the reviewers’ comments.

Please note: Line numbers changed due to our revision of the text and the inclusion of additional text recommended by the reviewers; hence line numbers do not coincide with the earlier version.

Reviewer 1

Open Review

English language and style

( ) Extensive editing of English language and style required 
(x) Moderate English changes required 
( ) English language and style are fine/minor spell check required 
( ) I don't feel qualified to judge about the English language and style 

Response: Thank you for your feedback, we carefully edit language.

Yes

Can be   improved

Must be   improved

Not   applicable

Does   the introduction provide sufficient background and include all relevant   references?

(   )

(   )

(x)

(   )

Is the   research design appropriate?

(   )

(x)

(   )

(   )

Are the   methods adequately described?

(   )

(   )

(x)

(   )

Are the   results clearly presented?

(   )

(   )

(x)

(   )

Are the   conclusions supported by the results?

(   )

(   )

(x)

(   )

Comments and Suggestions for Authors

Response: Thank you for suggestions, introduction has substantially improved, and methods described in detail. Minor edition has been made in the result and conclusion.

You have some useful and interesting data on tree growth patterns.  However, the current manuscript has almost no general context and thus is more appropriate for a regional than for an international journal.  I think that putting the study in a more general context is feasible, but this will require a substantial effort.  Also I have a number of other major concerns such as some procedures (e.g., sample selection) need to be more fully described, and there is some confusion or inconsistency in the presentation of results, which must be addressed.  Below I indicate my main points followed by a list of specific comments.

Response: Thank you we appreciate your suggestions and improved the introduction and try to make it more general with taking management intervention and also slightly revised the result.

 Main points:

 1.  The Introduction needs to be rewritten.  This section starts directly in with the species and the system.  However, for an international journal there needs to be an introduction to the general questions being addressed.  Otherwise the paper would be more appropriate in a regional journal.  The first two paragraphs of the Introduction would be better as part of the study area and species description; they could be incorporated into section 2.1.  The Introduction needs to start with a part on the relevance of the work to an international audience.  What are the issues of general relevance that are being addressed?

Response: Introduction are revised as per the suggestions.

2.  In the section on data collection (lines 161-171), it is essential that more information be given on sample selection.  How were trees chosen for sampling?  How many plots were used and how many trees were selected from each plot?  What was the stand structure and history of the forests in the plots?  Was there evidence of recent disturbance?  This kind of information is critical to interpreting patterns of growth of individual trees. 

Response: We revised and added few missing information.

3.  Given that you are working with a tropical species, it seems likely that rings could be indistinct and not necessarily indicate annual increments.  Thus, it is important to say something about the distinctness of rings and why you think they indicate annual increments.  Was there any evidence of false or missing rings, and are these likely to occur in this species?  These characteristics of the rings need to be described early on in the Methods.  They could be described at the beginning of section 2.3.2 where you address counting and measuring the rings. 

Response: Many thanks for your suggestions, we revisited according to your advice.

 4.  The trees sample all appear to have been very young, which is an important consideration for evaluating the results, especially since I assume this species can get very old, at least as indicated by the large size it can attain.  Early in the results there should be a good indication of the age of the trees used.  Sizes should also be clearly indicated as I am confused about the size range because it is indicated that the sizes are 10-50 cm DBH but in Figure 6 the largest tree is only about 24 cm DBH, and most trees are less than 20 cm DBH.  These inconsistencies need to be clarified. Also the sample sizes need to be clearly indicated for all results.

Response: In the forests, we can find the trees upto 50 cm DBH. We have collected the cut-stump samples from big trees as well (previous collections). However, inner portion of the bigger trees were rotten and hollow inside. That makes us difficult for proper identification of the tree age of old trees. Therefore, all bigger trees were discarded for measurement and further analysis in subsequent sections. It is clearly mentioned in the methodology and results

5.  Throughout the reporting of the results, there needs to be more care in clearly describing the results, especially in relation to the figures and tables.  For example on line 324 it is indicated that DBH and height are strongly correlated, but the panel on figure 6 shows no relationship at all between these variables.  The text of the results needs to be very carefully checked against what is actually presented in figures and table.  Also the figures need to be more clearly presented.  For example the axes labels on Figure 6 are way too small and the letters (a, b, and c) indicated in the legend are not visible on the figure.

Response: The  explanation of figure is added and figure 6 has been edited.

6.  There needs to be clarification of the nature of growth rings in this species and whether annual rings can be accurately determined and measured.  There are substantial inconsistencies in section 3 (Results and Discussion) concerning this issue.  For example, the statement on lines 465-468 would indicate much of the parts presented earlier, which are based on precise measurements of annual rings, are questionable.  These inconsistencies need to be fully explained and resolved.  As indicate in my third point above, you need to be very clear in addressing this issue early in the methods.  It is relevant to most of the analyses in the manuscript.

Response: The growth rings in the Shorea robusta are not very clear in general look as it is diffuse porous broadleaved hard wood tropical species. However, with proper sanding and polishing, and careful observation under the microscope it is distinct. To validate our ring boundary detection during counting, we did wood anatomical analysis and also followed the ring boundary detection criteria explained for S. robusta from Bangladesh in a paper published in journal Tree-Ring Research.

 7.   A critical concern is that no general discussion or conclusions are presented that would be of much relevance to an international audience.  Just as with the Introduction, in the last sections of the manuscript the study is placed only in the context of the study system and of the region where the study occurred.  It is essential that the work be placed in the context of concepts related to tree growth patterns and dendrochronology at a more general level.  I think that this is feasible, but has not been done.  Otherwise, the paper should be directed to a regional journal.    

Response: Thank you we substantially modified discussion and also added general paragraph at the end “In overall, our study is the one of the valuable tropical tree species revealed that there is a great prospect to extend dendrochronological research on tropical regions of the Indian sub-continent region to fulfill the research gaps. Dating of annually resolved tree-rings of tropical species implies that there is huge potential in tropical region to explore the forest growth, carbon sequestration, forest patch dynamics, climate- growth response analysis, and also to reconstruct past climate if we collect samples from old trees. Similarly, we can analyze age dependent growth and sensitivity to climate change and variability. In dense forest stands we can study the effects of completion on radial growth of trees. In those tropical tree species, in which the growth-ring may be asymmetrical in nature and localized growth in certain section of a trunk in some years, disc-sample will be more useful. In those cases where the collection of disc sample will not be possible, it will be better to collect more than two tree core samples from different direction of the tree for the accurate dating of trees. Trees in the tropical region can get benefit from a moist year in spring and summer with warm winter climate, especially in those moist tropical areas”. It is in line no 676-689.

Specific comments and suggestions for wording changes:

 Lines 26-29. This sentence is confusing and needs to be reworded.  

Response: It has revised.

Lines 29-30.  This sentence does not make sense without context provided.  Does this apply to individual trees or stands, and trees of what sizes and growing in what types of conditions?  These points are clearer later, but not here in the abstract.  It might be best to simply remove this sentence.    

Response: The sentence has been removed as suggested.

Line 35.  “the season” is unclear.  Which season?  Also this sentence seems to be rather redundant with the following sentence and perhaps is not necessary.     

Response: The sentence has been removed as suggested.

Line 56.  I do not understand the meaning of “from east to west”.    

Response: It has revised.

Lines 62-64.  Does this mean forests were the species is greater than 50% of the trees? 

Response: Yes! If other species dominate more than 50% then it is other hardwood forest’

Lines 69-71.  This part needs clarification.  Did these “naturally emerging” forests not have S. robusta before or do you mean the forests have a different structure because of altered management? 

Response: This sentence has revised.

Lines 80-84.  This part is confusing and seems out of place being attached to the end of a paragraph that mostly covers a different topic. 

Response: This has rearranged.

Line 110.  What does “appropriately” mean?  Do you mean correctly or accurately?    

Response: It refers for correctly.

Lines 114-117.  This sentence is awkward and needs to be reworded.

Response: It has revised.

 Lines 126-128.  This sentence is not necessary and can be deleted.

Response: Has been removed.

Lines 153-154.  The last part of this sentence is confusing.  What does “seedling decimation” mean and how does frost enrich moisture?

Response: Has been removed.

Line 174.  I suspect that “processed analyzed” should be “processed and analyzed”.

Response: Corrected as per your suggestions.

Lines 192-193.  How many samples were discarded?  I assume this reduces the number of samples used from the 60 indicated in section 2.2.  The final number of samples (trees?) used should be indicated here.

Response:  30 samples discarded.  30 cut-stump samples were used. As we have measured 4 radiii per tree, we have total 120 ring-width series. 30 trees have standard replicate for dendro research (fritts 1976, Speer 2010).

Lines 206-207.  This sentence is confusing.  What does “noise” mean here?

Response: In dendroclimatology or climate-growth relationship, our interest is to trace ‘climatic signal’ or influence of climate on growth. However, tree ring-width also contains biological or other non-climatic trends. For that non climatic trend we commonly term ‘noise’ for climatic response analysis which is corrected as well.

Line 241.  You need to clearly indicate what specific variables were used, especially what seasonal variables.  How many variables in total were used?  I suspect this is quite a few variables so was any adjustment made for significance of the correlation in relation to multiple comparisons?

Response: Clarified in text which is as follow and in line no 373-378.

-        Climatic variables includes average monthly temperature of 12 months starting  from January to December , monthly  total precipitation of 12 months starting  from January to December, and seasonal (spring = MAM or March-May; summer = JJAS or June-September; spring-summer = MAMJJAS or March-September; winter = NDJ or November-January) temperature and precipitation of Rampur stations.  Sometimes we combine seasonal data.

-        In general, total 15 temperature and 15 precipitation variables were used.

Line 256.  Were there still 60 samples after some were discarded, as indicated on lines 192-193?

Response: Sentence has been corrected.

Lines 257-260.  This part is confusing.  Are you indicating that there are missing or false rings, and do you mean by “synchronization” the number or the width of rings on different radii?  I assume width, but this needs to be clear here.

Response: Synchronization problem is not only due to pinching or missing rings, but also due to asymmetrical growth pattern of a ring throughout in the disc.  The “synchronization” here means matching the pattern of width of rings on different radii of a single tree first. Then, ring width pattern matching between the radii of different trees. We modified sentence for clarification.

Lines 270-271.  Does this mean that only 30 of the 60 trees were used?  Also were there any trees older than 28 years that were not included in the 30 trees?  What was the average age of the trees used?

Response: Yes, only 30 out of 60 trees were used. Yes, there were trees older than 28 years that were not included in the 30 trees.

Line 378.  How are young and mature age classes defined?  Because the oldest trees appear to be about 24 years old (Figure 6) can any of these trees have already reached a mature age class?

Response: Trees can be defined young and old looking their girth, structure, and age 9if available).  These trees are young, because we have found more than 50 years old trees for same species.

Lines 379-380.  You need to be cautious in extrapolating from individual tree to stand-level growth.  These can be very different, depending on stand characteristics such as tree size structure.  This distinction needs to be considered here and throughout the rest of this paragraph.

Response: Sentence has been corrected.

Line 400-404.  Global change could be important for growth of your trees, but you have no data to confirm this as young trees could be expected to grow rapidly in any case.  Thus caution in interpretation is warranted.

Response: Reviewer are right, but as we are removing the biological trend, we assume this will follow in older age tree too. But for generalization we don’t have data.

Lines 420-421.  This seems inconsistent with the indications above that growth rings were distinct and could be clearly measured.  This inconsistency is critical and must be explained.

Response: Sentence was rewritten for clarification. In general look, the rings are indistinct; however, with proper sanding and polishing of surfaces or suitable preparation of slides, they are distinguishable.

Line 486-488.  This type of comparison of the difficulty of determining growth rings in young versus older trees is not described earlier.  This comparison should not be in the Conclusions section if it is not fully described in the methods and results.

Response: Sentence modified for clarity.

Lines 491-493.  It is unclear what “enhanced” means here as this type of trend would be the normal expectation for young trees. 

Response: We had used ‘enhanced’ term to indicate increasing growth seen from chronologies. Now, ‘enhanced’ is removed.

Reviewer 2 Report

This ms provides an assessment of relationships between climate variables and radial growth of Shorea robusta. Authors make use of fairly standard dendrochronology methods.

The ms reads relatively well, but I did find an overabundance of details in multiple sections of the ms. Overall, I think the ms could be greatly improved by reducing the length of text, and present information in a more concise manner.

Data presentation in several tables and figures could also be improved significantly. Several section of Results seem to belong to Methods.

One notable comment is that authors provide no real recommendation for the management of the species, although they do mention the lack of updated management guidelines in the rationale for undertaling the study. Therefore, what does the study accomplish to achieve this goal? Definitely an area of improvement.

I provide several specific comments:

Abstract: shorten it; will be punchier if more concise

Line 19: “implications for forest management” – also mentioned in Introduction, but the ms provides, seemingly, zero implications for management in the Discussion. Who should care about this results (yes, line 500), but not quite clear why?

Line 20: “to predict growth” – I think the ms advances the understanding of climate effects on Sal’s growth, but I don’t see any actual contribution to prediction.

Line 35-36 – repetition of same ideas from previous phrase. Remove.

Line 42: “dendrometer” – like the measuring device? Maybe you mean ”dendrometrical data”, or better yet “Biometrical data”?

Lines 82-84: Apparently, something is missing from this long phrase.

Lines 85-97: This is general knowledge. Remove.

Lines 98-119: Please shorten, too long of a paragraph, makes it difficult to read.

Line 126: “Noble”? Might want to let readers decide on their own about the “nobleness” of this work

Line 141: “proposes” – purposes?

Lines 180-204: Shorten, shorten, shorten. Less details absolutely necessary.

Line 253: delete “to”?

Line 257: “dominant” – dominant in what sense exactly? Like high representation, or percentage? Dominant could also be a canopy social class.

Lines 255-286: This feels more like Methods than Results, including Fig. 3. Consider moving to previous section.

Lines 294-301: (including Table 1): Same as above, move to Methods.

Lines 308-313: This is general knowledge, consider removing.

Lines 314-317: Methods?

Line 413-414: In table 2: is BA per ha?

Discussion: Need to articulate what are the actual implications for management, even dare to suggest some possible recommendations?

Author Response

Manuscript: forests- 489388

Title: Dendroclimatological growth response of Shorea robusta using growth ring measurements

Thank you for providing the valuable suggestions and the opportunity to re-submit our manuscript. We thank both reviewers for their comments and suggestions. We believe that the following statements and the related changes in the manuscript would adequately address the reviewers’ comments.

Please note: Line numbers changed due to our revision of the text and the inclusion of additional text recommended by the reviewers; hence line numbers do not coincide with the earlier version.

Reviewer 2

Open Review

English language and style

( ) Extensive editing of English language and style required 
(x) Moderate English changes required 
( ) English language and style are fine/minor spell check required 
( ) I don't feel qualified to judge about the English language and style 

Response: We carefully checked the language again.

Yes

Can be   improved

Must be   improved

Not   applicable

Does   the introduction provide sufficient background and include all relevant   references?

(   )

(x)

(   )

(   )

Is the   research design appropriate?

(x)

(   )

(   )

(   )

Are the   methods adequately described?

(   )

(x)

(   )

(   )

Are the   results clearly presented?

(   )

(   )

(x)

(   )

Are the   conclusions supported by the results?

(   )

(   )

(x)

(   )

Comments and Suggestions for Authors

This ms provides an assessment of relationships between climate variables and radial growth of Shorea robusta. Authors make use of fairly standard dendrochronology methods.

Response: Thank you for your suggestion, research design and methodology has slightly improved and result and conclusions has improved.

The ms reads relatively well, but I did find an overabundance of details in multiple sections of the ms. Overall, I think the ms could be greatly improved by reducing the length of text, and present information in a more concise manner.

Response: Thank you for your suggestion, we tried our best to make MS more concise.

 Data presentation in several tables and figures could also be improved significantly. Several section of Results seem to belong to Methods.

Response: Figure 6 has improved.

One notable comment is that authors provide no real recommendation for the management of the species, although they do mention the lack of updated management guidelines in the rationale for undertaling the study. Therefore, what does the study accomplish to achieve this goal? Definitely an area of improvement.

Response: Thank you for your suggestion, we added in the conclusion section.

Specific comments:

Abstract: shorten it; will be punchier if more concise

Response: Thank you for your suggestion, tried to make more concise.

Line 19: “implications for forest management” – also mentioned in Introduction, but the ms provides, seemingly, zero implications for management in the Discussion. Who should care about this results (yes, line 500), but not quite clear why?

Response: Many thanks we added the management implication in introduction and conclusion sections.  

Line 20: “to predict growth” – I think the ms advances the understanding of climate effects on Sal’s growth, but I don’t see any actual contribution to prediction.

Response: We have modified accordingly.

Line 35-36 – repetition of same ideas from previous phrase. Remove.

Response: Removed.

Line 42: “dendrometer” – like the measuring device? Maybe you mean ”dendrometrical data”, or better yet “Biometrical data”?

Response: We are talking about the dendrometer based continuous growth measurement. Dendrometer could be manual band dendrometer or automated digital dendrometer.

Lines 82-84: Apparently, something is missing from this long phrase.

Response: Corrected

Lines 85-97: This is general knowledge. Remove.

Response: Shortened and modified.

Lines 98-119: Please shorten, too long of a paragraph, makes it difficult to read.

Response: Shortened and modified.

Line 126: “Noble”? Might want to let readers decide on their own about the “nobleness” of this work

Response: Modified.

Line 141: “proposes” – purposes?

Response: Corrected.

Lines 180-204: Shorten, shorten, shorten. Less details absolutely necessary.

Response: Done. We tried to make concise.

Line 253: delete “to”?

Response: Done

Line 257: “dominant” – dominant in what sense exactly? Like high representation, or percentage? Dominant could also be a canopy social class.

Response: Corrected as frequent.

 Lines 255-286: This feels more like Methods than Results, including Fig. 3. Consider moving to previous section.

Response: Moved to methods section as per suggestion.

Lines 294-301: (including Table 1): Same as above, move to Methods.

Response: Table was removed, and results were directly explained in the text.

Lines 308-313: This is general knowledge, consider removing.

Response: Considering the broader audience of journal, we had explained in details

Lines 314-317: Methods?

Response: Modified

Line 413-414: In table 2: is BA per ha?

Response: Yes, corrected.

Discussion: Need to articulate what are the actual implications for management, even dare to suggest some possible recommendations?

Response: Discussion added as per suggestion.

Round 2

Reviewer 1 Report

I reviewed your manuscript previously, at which time I had some substantial concerns.  You have effectively addressed these concerns in your revision.  The revision reads well in general and effectively presents your study.  I see no need for further substantial revision.  I think that you have conducted a useful study that will be a nice contribution to the literature on the use of tree rings to examine patterns of tree growth.

Author Response

Reviewer 1: 

There isn't any comments and suggestions. Thank you for your appreciation.

Reviewer 2 Report

Authors seem to have considered most of the suggestions provided in the initial evaluation by reviewers. While some sections have been shortened or removed, the revised ms is still slightly longer than the initial version. I have one concern left: authors talk at length about management implications in the Introduction. However, very little, if any, has been provided in the Discussion: what does this study provide for management? Any recommendations that could be made? I don’t think it is enough to say that dendrochronology holds a lot of potential for several domains. Alternatively, if authors do not have anything to offer as management implications from this study, then the management context should be removed from the Introduction

Author Response

Reviewers 2: Many thanks for your suggestions;

Authors seem to have considered most of the suggestions provided in the initial evaluation by reviewers. While some sections have been shortened or removed, the revised ms is still slightly longer than the initial version.

Response: We have revised and tried to make more shorten.

I have one concern left: authors talk at length about management implications in the Introduction. However, very little, if any, has been provided in the Discussion: what does this study provide for management? Any recommendations that could be made? I don’t think it is enough to say that dendrochronology holds a lot of potential for several domains. Alternatively, if authors do not have anything to offer as management implications from this study, then the management context should be removed from the Introduction

Response:  Many thanks for pointing out, as suggested we revised where are when  needed in whole the manuscript. We workout in the introduction, result,  discussion and conclusion and added management perspectives in  discussion.